# The Cardiac Effects of Performance-Enhancing Medications: Caffeine vs. Anabolic Androgenic Steroids

**DOI:** 10.3390/diagnostics11020324

**Published:** 2021-02-17

**Authors:** Sanjay Sivalokanathan, Łukasz A. Małek, Aneil Malhotra

**Affiliations:** 1Cardiovascular Clinical Academic Group, St. George’s University of London and St. George’s University Hospitals NHS Foundation Trust, London SW17 0RE, UK; ssivalok@sgul.ac.uk; 2Department of Epidemiology, Cardiovascular Disease Prevention and Health Promotion, National Institute of Cardiology, 04-628 Warsaw, Poland; lmalek@ikard.pl; 3Division of Cardiovascular Sciences, University of Manchester and Manchester University NHS Foundation Trust, Manchester Institute of Health and Performance, Manchester M11 3BS, UK

**Keywords:** sports cardiology, athlete, caffeine, anabolic androgenic steroids, heart disease, cardiac magnetic resonance imaging

## Abstract

Several performance-enhancing or ergogenic drugs have been linked to both significant adverse cardiovascular effects and increased cardiovascular risk. Even with increased scrutiny on the governance of performance-enhancing drugs (PEDs) in professional sport and heightened awareness of the associated cardiovascular risk, there are some who are prepared to risk their use to gain competitive advantage. Caffeine is the most commonly consumed drug in the world and its ergogenic properties have been reported for decades. Thus, the removal of caffeine from the World Anti-Doping Agency (WADA) list of banned substances, in 2004, has naturally led to an exponential rise in its use amongst athletes. The response to caffeine is complex and influenced by both genetic and environmental factors. Whilst the evidence may be equivocal, the ability of an athlete to train longer or at a greater power output cannot be overlooked. Furthermore, its impact on the myocardium remains unanswered. In contrast, anabolic androgenic steroids are recognised PEDs that improve athletic performance, increase muscle growth and suppress fatigue. Their use, however, comes at a cost, afflicting the individual with several side effects, including those that are detrimental to the cardiovascular system. This review addresses the effects of the two commonest PEDs, one legal, the other prohibited, and their respective effects on the heart, as well as the challenge in defining its long-term implications.

## 1. Introduction

Caffeine (1,3,7-Trimethylxanthine) is a popular workplace substance that has been well-researched, with its ergogenic effects being known for centuries [1]. Caffeine has a wide range of acute benefits that includes an increase in alertness and concentration, accompanied by a reduction in fatigue and pain perception [2,3]. As a result, its use has become highly prevalent amongst athletes, especially after 2004, when it was removed from the World Anti-Doping Agency (WADA) list of banned substances; it was, therefore unsurprising when a study reported that 74% of urine samples from athletes, between 2004 to 2008, demonstrated measurable levels [1]. Common physiological effects of caffeine on the body include an increase in heart rate, catecholamine levels, blood lactate, free fatty acids and glycerol [4]. More significantly, its use has illustrated benefits in both endurance-based and high-intensity exercise, permitting the athlete to train longer and at a greater intensity. A recent meta-analysis yielded a positive relationship of caffeine on muscle strength, muscle endurance and anaerobic power [5]. As a result, it is recommended that ingestion of 3–9 mg/kg approximately 60 min prior to exercise may provide the extra competitive advantage for the athlete [1]. Nonetheless, the response to caffeine is multifaceted, influenced by both genetic and non-genetic predilections, with there being inter-subject variation in response to caffeine consumption, and this heterogeneous response makes it difficult to extrapolate the objective impact of caffeine as a vital ingredient to athletic prowess.

In contrast, anabolic androgenic steroids (AASs), synthetic derivatives of testosterone, have been abused by athletes since the 1950s for their ability to increase muscle mass and improve athletic performance. The terms anabolic and androgenic refer to muscle hypertrophy and increased male sex characteristics, respectively. AASs are artificial substances that act on androgen receptors and are commonly used in the treatment of metabolic or catabolic disorders and other chronic conditions related to low testosterone [6]. More significantly, its misuse stems from the means of achieving a lean and muscular body type, with the potential of shielding the user from muscle fibre damage, through enhanced protein synthesis during recovery. There are multiple manufactured forms, most of which are designed to optimise muscle growth whilst minimising the undesired androgenic effects [6]. Steroid abuse has dramatically increased over the past two decades in the general population who live in an increasingly image-obsessed era. Its users are typically 20–30-year-old males, who participate in recreational exercise largely composed of weight training [6]. Globally, it is estimated that 6.4% of males and 1.6% of females use AASs [7]. The second highest prevalence of users beyond recreational sportspeople (18.4%) are athletes (13.4%) [8]. Whilst anabolic androgenic steroids can play an important role in clinical treatment of endocrine disorders there are several established adverse outcomes, if misused, that includes an increased risk of cardiovascular disease (CVD), risk of tendon ruptures, hepatorenal disorders and psychiatric symptoms. The doses are often 5–15 times higher than recommended levels, with athletes experiencing a higher probability of adverse cardiovascular events that includes stroke and myocardial infarction (MI) [9]. Preceding these events are hypertension and left ventricular (LV) hypertrophy, both independent predictors of cardiovascular mortality and morbidity [10,11]. There are, however, many obstacles to the investigation of the dangers of AASs, due to the dose never being reliably known, to polypharmacy or the ethical restrictions of conducting research studies [6].

Given such a variability in effects of both caffeine and AASs, this review discusses the impact of the two commonest performance-enhancing drugs (PEDs) and its documented cardiac sequalae.

## 2. Materials and Methods

We performed a comprehensive search on Pubmed, and Scopus focusing on the effects of caffeine and/or AASs to exercise and its subsequent effects on the myocardium (Appendix A
Figure A1). Reviews, meta-analyses, prospective, retrospective, interventional and observational studies were included in our search. Exclusion criteria included conference abstracts, or articles where correlation between exercise, the cardiovascular system (CVS) and caffeine or AASs did not exist. The review of AASs was limited to findings after the year of 1986, as widespread testing became available in Europe and the United States at the end of 1986. Key search terms included: “caffeine”, “caffeinated”, “CAF”, “tea”, “energy drinks”, “anabolic androgenic steroids” in combination with “exercise”, “athlete”, “myocardium”, “cardiac”, and “heart”.

## 3. Results

### 3.1. Caffeine as a Performance Enhancing Agent

In many sports, changes in performance of 1% may be the difference between first or second place [12]. Caffeine is a readily available performance enhancing aid that improves athletic ability across virtually all sporting disciplines. Historically, it was recommended to be banned in 1939, due to its ergogenic properties that may influence sporting accomplishments. Since its legalisation in 2004, it has become a major source for athletes, commonly being in the form of energy drinks, but may vary in the form of a gum, gel, pill or inhaler. Through fat mobilisation and thus sparing of the glycogen reserve, it diminishes the impact of fatigue, pain and effort that is associated with exercise, leading to the more significant motives of athletes for its consumption. A typical 250 mL energy drink (ED) may contain up to 80 mg of caffeine, similar to that in filtered coffee (90 mg), and twice the amount of that in tea (30 mg); additional substances that complement the influence of caffeine include ginseng, taurine and guarana [13,14,15,16,17].

Caffeine use may be classified as low, moderate or high, with ingestion of ~3 mg/kg (~200 mg for a 70 kg individual; 1–2 small cups of coffee) being considered low, 5–6 mg/kg considered moderate and ~10–13 mg/kg viewed as high [17]. It should be noted that the dose-response relationship between caffeine and athletic performance has yet to be established, with low dose caffeine appearing to exhibit the most ergogenic effect on athletes. For instance, caffeine containing drinks, with a dose equivalence to 3 mg/kg, have shown an increased ability of football players in sprinting, jumping and the distance covered [18]. Further meta-analyses investigating the role of caffeine have demonstrated a significant increase in jump height, muscular endurance, aerobic endurance performance and muscle strength [19].

Like most substances, caffeine, when consumed in larger doses, may result in side effects that includes dehydration, seizures, migraines, insomnia, arrhythmias, gastrointestinal problems and psychological permutations [15,16,20].

#### 3.1.1. Caffeine Pharmacology and Cardiac Physiology

Caffeine is rapidly absorbed by the body, with its concentration peaking between 40 to 80 min, and rising to ~15–20 µmol/L with a low caffeine dose, ~40 µmol/L with a moderate and ~60–70 µmol/L with a high dose. It appears in the blood within 5–15 min of ingestion and has a long half-life (3–5 h) [21]. For both female and male athletes, for a given dose of caffeine, it appears that the concentration of caffeine and its metabolites are the same [22,23].

The effects of caffeine are exerted primarily through the blockade of adenosine receptors (subtypes A_1_ and A_2_), which are found throughout the myocardium and coronary circulation; they are also found in the brain, adipocytes, skeletal and smooth muscle (Figure 1). The result in the competitive blockade of these receptors leads to an increase in peripheral vascular resistance, sympathetic tone and increase in renin, subsequently amplifying the heart rate, cardiac contractility and blood pressure [24]. Secondary metabolic changes of caffeine include stimulating the secretion of epinephrine.

Whilst the concerns of caffeine on overall health has permeated through society, there are many epidemiological studies that have shown its benefit to overall mortality, and in particular cardiac disease [25]. Moreover, although historical studies have demonstrated an increased risk in MI and CVD [26,27], a study of 45,589 men and 85,747 women followed up for 2 and 10 years, respectively, did not show a substantial risk in CVD [28]. Caffeine may, however, conversely attenuate the physiological response to exercise, such that there may be reduced coronary blood flow or response of the endothelial cell in mediating the vascular tone during exercise, which signifies a potential risk to an athlete with silent coronary disease. Other impacts of caffeine include a delayed return of the parasympathetic nervous system, and with a state of sustained sympathetic activity, this may confer an increased risk of life-threatening arrhythmias [21].

#### 3.1.2. Caffeine and Risk of Arrhythmia

Whilst many studies have reported the arrhythmogenic effect of caffeine, it has not been replicated on large population studies. With the consumption of caffeine being ubiquitous in Western society, the widely held belief that caffeine may contribute to arrhythmia or the risk and development of coronary heart disease may not be evidence-based [24,25,29,30,31,32]. Intoxication of caffeine, however, is still reported, demonstrating its potential in provoking fatal arrhythmias [33]. Physiologically, through the blockade of calcium reuptake into the sarcoplasmic reticulum, and thus a rise in intracellular calcium, the potential of atrial arrhythmia, through enhanced automaticity of atrial pacemaker cells, exists; three cups of coffee (250 mg) have shown to increase both epi- and norepinephrine [34]. More importantly, energy drinks often contain caffeine at a significantly higher concentration than either coffee or tea; the stimulant properties of other compounds in EDs, such as taurine, complicates matters further. Taurine, for instance, is suggested to increase calcium accumulation in the sarcoplasmic reticulum, favouring the excitation-contraction of skeletal muscles, but may also induce unfavourable arrhythmias [35].

It could be argued that the absence of risk may not relate to athletes or those who harbour an underlying abnormal cardiac substrate, especially as the amount of caffeine consumed through energy drinks may be invariably higher. For instance, there has been reports of EDs prolonging QTc and unmasking Brugada syndrome [34]. Another important impact of caffeine includes the augmentation of ryanodine receptors, that may further lead to an increase in calcium release within cardiac cells, affecting the heart’s ability to contract and use oxygen, which may predispose to arrhythmias [36].

On the other hand, when attempting to explore the relationship between caffeine and arrhythmias in those with pre-existing cardiac disease, there failed to be a connection, suggesting the complex pharmacodynamics of caffeine [33].

#### 3.1.3. Caffeine Genetics

It is evident that genetic factors demonstrate a huge role on the individual response to the effects of caffeine [37,38,39]. Whilst its mechanisms may not be well defined, there are certain drivers of these individual differences; notable genes include CYP1A2, ADORA2A and catechol-*O*-methyltransferase (COMT) [40]. Of the most significance is CYP1A2, which is involved in the breakdown of caffeine and has two alleles (A & C), dichotomising into either fast or slow metabolisers, respectively. The significance of this phenomenon is that those who are slow metabolisers, who consume moderate (3–4 cups) amounts of coffee have a greater risk of hypertension and MI [1]. This is also reflected in athletes, with those who are fast metabolisers showing greater improvement in performance; this may be due a rapid accumulation of caffeine metabolites, and may reflect why timing of caffeine consumption becomes important [1].

In contrast, polymorphisms affecting ADORA2A could lead to an individual to experience greater sleep disturbance, impacting athletes that compete in the evening, or increased anxiety resulting in poor competition performance [12].

#### 3.1.4. Caffeine in Sudden Cardiac Death

Sudden cardiac death (SCD) is defined as an unexpected death or arrest, presumed to be secondary to a cardiac cause, within 1 hour of symptoms or, if unwitnessed, within 24. Energy drinks has been associated with coronary vasospasm and ischaemia, arrhythmias, endothelial dysfunction and increased platelet aggregation [41]. Its use has been a particular concern amongst the younger athletes, where case reports of sudden cardiac death were in part attributed to the consumption of energy drinks [41]. However, whilst no direct link between caffeine and its supposed harmful effects on the heart exist, further studies are required to establish its true safety, particularly in those with underlying electrical or structural cardiac abnormalities. Additional studies would be important in recognising the effects of strength and delivery of caffeine, and the effects of age and genetic expression on the individual’s response to caffeine.

### 3.2. Anabolic Androgenic Steroids as a Performance Enhancing Agent

Anabolic androgenic steroids first gained popularity in the 1954 Olympics and given its potential to improve physical ability, appearance and performance, it has been banned for any sporting use since 1974. Regardless, it is continued to be misused by athletes in sports such as weightlifting, football, cycling and many others to improve both performance and in order to gain a competitive advantage; it is reported that up to 50% of positive doping cases account for AAS use [42]. The lifetime prevalence of AASs ranges from 1–5% in Western countries, and its use has increased four-fold, since 2016, from 0.1% to 0.4% of the population, affecting an extra 19,000 young people (aged 16–24 years old). Although AASs are commonly administered subcutaneously or intramuscularly, it may also be delivered as oral or transdermal preparations or as an implant [43]. Motivators for its use include achieving rapid muscle growth, greater than can be achieved by exercise alone [6]. It can often be problematic to attribute the harmful cardiac effects of AASs, as users often take other compounds such as ephedrine, growth hormone, thyroxine and amphetamines [7,43]. Nonetheless, with mounting evidence in developing several physical and psychological health disorders, its use has become more than a concern restricted to athletes but one of public health.

#### 3.2.1. AASs Pharmacology and Cardiac Physiology

Anabolic androgenic steroids upregulate and increase the number of androgen receptors, increasing the transcription of DNA in skeletal muscle required for muscle growth, thereby contributing to an increase in muscle size and strength. It also includes a direct effect on cardiac muscle metabolism, altering both electrical and structural features of the myocardium [44]. Supraphysiological doses of AASs induces toxicity of the CVS, with the proposed mechanisms including changes in the lipid profile, elevations in blood pressure, myocyte hypertrophy, disarray and apoptosis and a procoagulant state [45]. Thereby, contributing to disorders such coronary artery disease (CAD), hypertension, cardiomyopathy and thromboembolic disorders (Figure 2); the above findings have been correlated with histopathological case reports [46,47]. Physiological changes include alterations in the lipid profile that includes a reduction (up to 20%) in high density lipoprotein (HDL), an increase (up to 20%) in low-density lipoprotein (LDL) and an increase in total cholesterol levels, which is accompanied with an increase in HMG-CoA reductase enzymes [6]. Such changes in lipid characteristics increases the hazard of CAD by 3–6 fold and may occur as quickly as 9 weeks since the onset of AAS use [48]. Hypertension, another commonly reported phenomenon in AAS users, is described to be a consequence of increased sympathetic drive and endothelial dysfunction [6]. The progression of such events is often hard to define, attributed to both dose and drug duration, but some are argued to be non-reversible, resulting in those to require cardiac devices or listed for transplantation.

AASs are involved in promoting the growth of cardiac tissue, resulting in significant adverse adaptations such as an increase in wall thickness, and left ventricular cavity size; there has been observable differences in left ventricular posterior wall and septal wall thickness [49]. The induction of myocyte hypertrophy results in counter opposing measures such as the release of apoptogenic factors leading to further deleterious effects on the myocardium (Figure 3). For instance, it has been noted that AAS abusers demonstrate a reduction in peak strain and strain rates of the left posterior and septal walls [50]. Diastolic function also appears to be affected, whereby a reduction in early and late diastolic filling velocity ratios is expected; a reduction in myocardial relaxation through increased collage cross-linking and fibrosis may explain such a phenomenon in anabolic androgenic steroid use [51]. Animal models have been particularly useful in demonstrating such changes. For instance, rats after 8–12 weeks of AAS use demonstrated cardiomegaly [45]. Furthermore, immunohistochemical analyses revealed greater expression of TNF-α and IL-1β (proinflammatory mediators), signifying ongoing silent myocardial injury in AAS users [52]. Post-mortem studies have also demonstrated adverse phenotypical changes to AASs such as cardiomegaly, myocardial fibrosis and necrosis [49]. Other ramifications include an increase in ventricular rigidity, as its use may reduce myocardial compliance through an apoptogenic effect on the cardiac myocytes [53]. More importantly, the effects of AASs are not limited to the left ventricle and several studies have suggested a global impact. For instance, there is an increase in right ventricular strain, and left atrial dysfunction [54]. As a result, AASs have led to the emergence of acquired cardiac disease in younger and middle-aged athletes.

Other important manifestations of anabolic androgenic steroid abuse include myocardial infarction and heart failure, secondary to premature atherosclerosis; infarcts may even occur without significant coronary vessel disease [55]. Animal models have illustrated increased androgen-induced vascular calcification, which could be secondary to steroid induced cell damage resulting in loss of tissue elasticity and thus fibrosis [56]. A landmark study among experienced male weightlifters reported that long-term AASs use was associated with myocardial dysfunction and accelerated coronary atherosclerosis [51]. Stroke is a particular risk with AAS use, with current guidelines advocating against the use of testosterone in patients who have experienced MI or stroke within the last 6 months [9]. The proposed mechanisms include hyperaggregation of platelets, increased plasma levels of factor VIII and IX, and heightened fibrinolytic activity through increased tissue plasminogen activator (t-PA) levels. Moreover, AASs can promote polycythaemia, through increased red cell production, leading to potential ischaemic events [9]. These forms of anabolic androgenic steroid-associated adverse cardiovascular phenotypes may represent a previously underrecognized public-health problem.

#### 3.2.2. AASs and Risk of Arrhythmia

Several studies have illustrated how the supraphysiological doses of AASs induces both morphological and electrical ventricular remodelling that results in cardiac autonomic dysfunction [57]. More importantly, hypertrophy, fibrosis and necrosis, repercussions of AAS use, are substrates for arrhythmias that are further compounded by exercise. Testosterone, in particular, has been associated with rhythmic disturbances, possibly through the potentiation of potassium channels involved in ventricular repolarisation, which could explain the presence of QRS-wave delay, sinus tachycardia and supra- and ventricular arrhythmias [44,58,59]. Signal-averaging electrocardiography (SAECG), a method of distinguishing conduction abnormalities has revealed longer QTc interval and QT dispersion in AAS users. Subsequently, increasing the likelihood of abnormal rhythms and SCD after or during exercise [58,59].

#### 3.2.3. AASs Genetics

Interindividual variation in genetics exist and alterations in cytochrome P450 (CYP450) and uridine diphosphate glucuronosyltransferase (UGT) enzymes may explain why certain individuals may require greater amounts of AASs or experience the more harmful effects. To date, no studies have evaluated whether genetic variation in AAS users play a role in the predilection of CVD. There has, however, been studies on the overexpression of molecular mediators, argued to be drivers of CVD. This includes overexpression of calcium/calmodulin dependent protein kinase II delta (CaMKIIδ), beta myosin heavy chain (MHC) and monoamine oxidase (MAO), hallmarks of pathological changes within the myocardium, such as myocyte apoptosis, cardiac hypertrophy, slow shortening velocity of cardiac fibres and arrhythmias [60]. More significantly, imbalance of Ca^2+^ homeostasis and increased CaMKIIδ activity is observed in both human and animal models of heart failure [60]. The overproduction of MAO is also related to ventricular dysfunction, apoptosis and fibrosis [60].

#### 3.2.4. AASs in Sudden Cardiac Death

Anabolic androgenic steroids have the potential of increasing the risk of sudden cardiac death through multiple mechanisms; it is, unfortunately, unclear to the exact nature of these events, especially since those who misuse often use a combination of drugs. Structurally, several modalities of dysfunction that may predispose to sudden cardiac death have been proposed that includes increased coronary artery plaque volume, cardiac hypertrophy, ventricular dilatation, myocardial fibrosis and cardiomyopathy [45,51]. Precisely, four mechanisms have been postulated to describe SCD in AAS users, that includes the thrombosis model, the atherogenic model, the direct myocardial injury model, and model of vasospasm [43]. The thrombosis model suggests that there is an increased risk in thrombus formation as a result of polycythaemia and increased platelet generation and aggregation. More importantly, with ongoing endothelial dysfunction, it precludes AAS users to fatal thrombotic complications, such as ischaemic stroke or pulmonary embolism. Additional physiological changes include the promotion of thromboxane A2 and thrombin, inducing a state of hypercoagulability [45]. In the atherogenic model, the heightened risk in SCD is accounted by the changes in the lipid profile leading to premature atherosclerosis and thus myocardial infarction; the study by Pärssinen accounted as high as 38% to MI [61]. The vasospastic model explains the occurrence of infarction, as a consequence to coronary vasospasm from the release of nitric oxide, in those with no evidence of atherosclerosis or coronary thrombosis [61,62]. The final model describes fatal arrhythmias precipitated by the chronic ischaemic damage brought about by apoptosis, collagen deposition and microcirculatory disturbance [62]. Suggested mechanisms that induces arrhythmias include increased QT interval, Tp-e interval, Tp-e/QT ratio and Tp-e/QTc ratio [45,59]. Furthermore, there have been several case reports of young athletes that have developed rapidly progressive (dilated) cardiomyopathy [45]. Isolated ventricular arrhythmias are another possibility, where it has been suggested that AASs inhibit the re-uptake of catecholamines, and with a combination of exercise, stimulating the nervous system that may increase the likelihood of fatal arrhythmias.

## 4. Discussion

Our findings suggest that whilst caffeine does not have noticeable structural changes on the myocardium, AASs has several. For instance, imaging and histopathological samples have demonstrated left ventricular hypertrophy, cardiomegaly and interstitial fibrosis, respectively [43]. Such remodelling has ramifications on the CVS, not only immediately but in the long-term as well. Substantial cardiovascular changes include increase in vascular tone and elevation in blood pressure, alterations in lipid profile and direct myocardial toxicity, resulting in reduced left ventricular function, cardiac hypertrophy and arterial and venous thrombosis [43,63]. In contrast, there is a lack of compelling evidence to suggest that caffeine has lasting morphological changes to the myocardium (Figure 4).

Arrhythmias have also been reported in individuals that consume caffeine or use AASs. Whilst the mechanisms have been discussed, and at length, it is often difficult to reproduce the results in larger studies. In terms of SCD, whilst evidence leans towards AASs, it is less apparent for caffeine. Direct effects of AASs include ventricular remodelling leading to cardiomyopathy. More importantly, left ventricular hypertrophy and fibrosis have both been identified as risk factors to SCD [64,65], and therefore may be argued to be the sequelae of ongoing AASs use that results in the terminal event. More precisely, ventricular arrhythmias, brought about by changes in the myocytes, interstitium and coronary flow reserve could lead to the fatal event [43]. However, it is not uncommon for AASs to stack or combine with other illicit drugs, making it a challenge whether such modifications are solely attributable to AASs. Furthermore, with the presence of physiological remodelling to exercise and thus the presence of the “athlete’s heart”, that includes left ventricular hypertrophy and increased cavity dimensions, it can be difficult to delineate whether the changes are brought about by AAS use or through exercise. Therefore, detraining could be an option if there is uncertainty to the aetiology.

Ultimately, to negate the suggested adverse cardiovascular effects, both the subject and healthcare professional has to take into consideration the risk of withdrawal if stopped abruptly. Unfavourable symptoms such as headache and irritability would naturally deter the individual in abstaining from caffeine. More importantly, in AAS use, hypogonadism and depression are durable side effects that prevent the user from refraining in its use. Furthermore, it is possible that despite discontinuation, there may be permanent changes from AASs that includes insulin resistance, hypertension and visceral adipose tissue [60].

### Clinical Pitfalls and Future Directions

Even though certain physiological mechanisms have been argued to be the benefits or drawbacks of both caffeine and anabolic androgenic steroids, these are still viewed as hypotheses and associations and, thus remain incomplete as explanations without larger randomised controlled trials. To what extent caffeine may be regarded as a drug to the athlete is difficult. There are several preparations, majority of which are in combinations, and there is lack of consistency on both performance and cardiac outcomes. Furthermore, it should be noted that the documented adverse effects of AASs have failed to be replicated in a few studies. For instance, not all AAS users experience left ventricular hypertrophy and/or endothelial dysfunction. Such effects could be explained by the use of allied substances or stimulants, such as cocaine, that are harmful to the CVS. In addition, AAS users often combine different forms of steroids, termed “stacking”, as a means of maximising muscle growth, which could account for the adverse cardiac effects experienced in some and not others [6]. More importantly, the varying structure, metabolites and administration patterns of AASs makes it challenging to predict accurate and reproducible physiological consequences on the cardiovascular system. Additionally, the majority of studies are focussed on post-adolescent males, becoming problematic when translating the negative cardiac effects to all users.

Furthermore, the discrepancies observed may depend on the age, sex, coexisting clinical conditions and status of athletic performance. As such this topic requires further studies in different clinical and sport settings. From a methodological point of view, in clinical studies on athletes, it is usually not straightforward to fully confirm or exclude the use of doping agents, which is a serious confounder. Future research may include the use of new promising biomarkers such as microRNAs [66,67]. Those small particles regulate the post-transcription gene expression by RNA-RNA interactions. Circulating microRNAs, due to their high environmental stability and presence in various body fluids, have been already shown to have potential in detection of illicit substances [43]. Several microRNAs have been also linked to heart dysfunction in the form of myocardial ischemia, hypertrophy, fibrosis and arrhythmia [66]. However, many of the same microRNAs also become up-or down- regulated in response to exercise as demonstrated in a recent review [67]. Therefore, their potential future use will depend on the ability to distinguish physiological adaptive changes to exercise from changes related to the use of illicit drugs.

## 5. Conclusions

There is a large growing body of evidence that describes the impact of both caffeine and anabolic androgenic steroid use on the cardiovascular health of both the athlete and non-athlete. Whilst caffeine may not necessarily give an athlete the essential edge, its use may not disadvantage them either, especially since the majority have consumed such a supplement prior their sporting event. In contrast, AASs have documented improvement in athletic proficiency. However, it does not negate the several adverse cardiovascular effects that is associated with its use. With the continued use of both caffeine and AASs, regular assessment, that includes evaluating the electrical activity and morphology of the myocardium, using non-invasive imaging and functional methods would be important in identifying those who are at an increased risk of cardiovascular disease or an acute cardiac event.

## Figures and Tables

**Figure 1 diagnostics-11-00324-f001:**
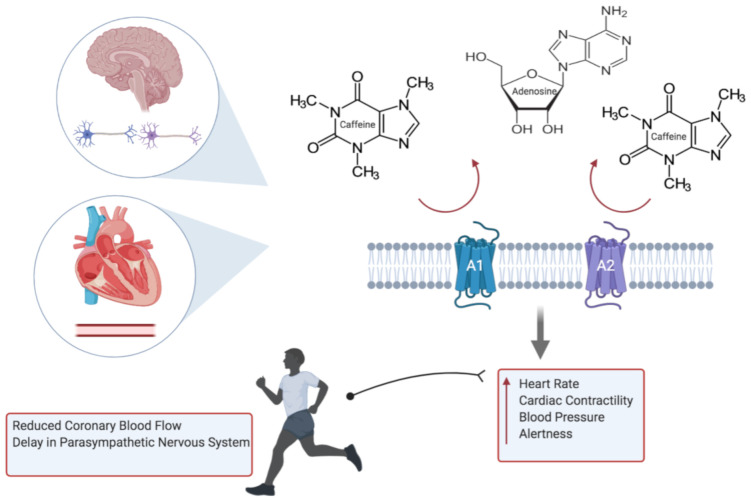
Caffeine inhibits the action of adenosine through the blockade of A_1_ and A_2_ receptors, resulting in elevated heart rate, blood pressure, cardiac contractility and alertness. Subsequent adverse cardiovascular events during exercise include potentiation of hypoxic damage to cardiac myocytes, through failure in relaxation of the coronary vessels, and arrhythmias (created with BioRender.com).

**Figure 2 diagnostics-11-00324-f002:**
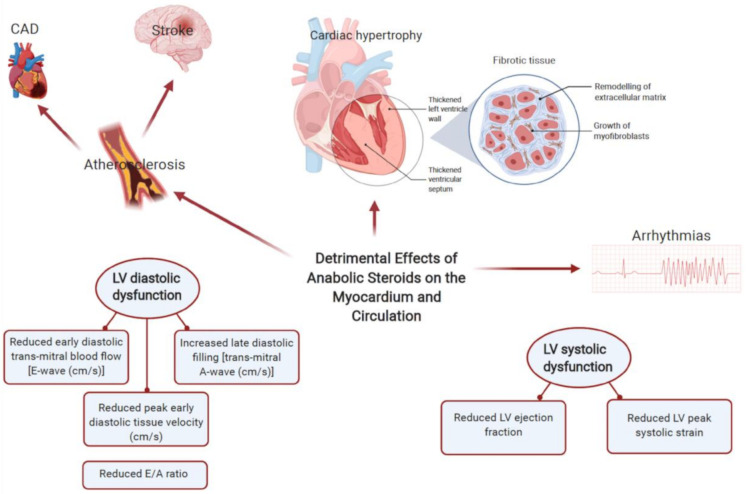
Common adverse cardiovascular effects of anabolic androgenic steroid abuse include vascular calcification, accelerated atherosclerosis, myocardial apoptosis, cardiac hypertrophy and arrhythmias. Impaired LV relaxation is a cardinal feature of the adverse cardiac effects of anabolic androgenic steroids (AASs). With long term abuse, there is evidence of reduced systolic strain and systolic dysfunction with resultant cardiomyopathies. Other sequalae of AAS abuse include increased incidence of thromboembolism and hypertension (created with BioRender.com).

**Figure 3 diagnostics-11-00324-f003:**
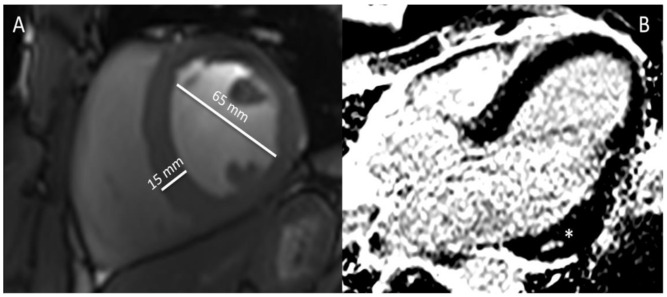
(**A**,**B**). Cardiac magnetic resonance (CMR) images of a 38-year old bodybuilder with anabolic androgenic steroid use—(**A**). Cine steady-state free precession (SSFP) in mid-ventricular short-axis view at end-diastole showing hypertrophied interventricular septum (15 mm) and enlarged left ventricle (62 mm) with decreased systolic function (ejection fracTable 44. not shown), (**B**). Late gadolinium enhancement (LGE) image in 3-chamber view showing midventricular area of fibrosis (non-ischemic) in the basal infero-lateral segment of the left ventricle (asterisk).

**Figure 4 diagnostics-11-00324-f004:**
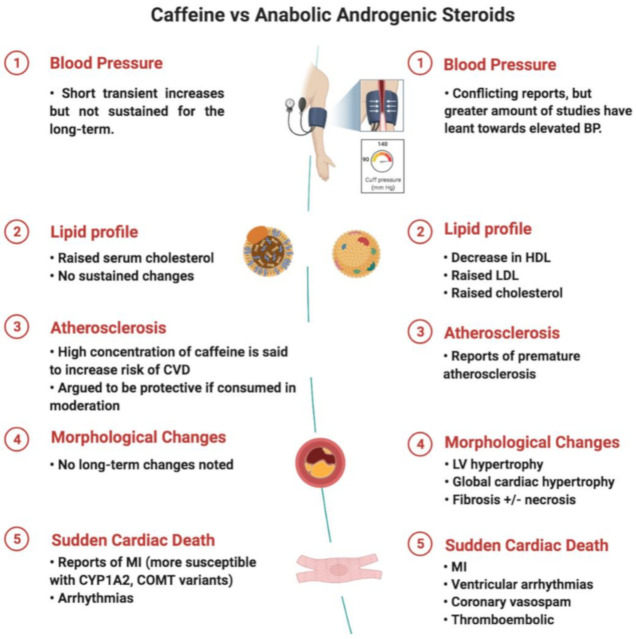
Comparison between caffeine and AASs and its associated cardiovascular effects (created with BioRender.com).

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
