# Peer review of "The Cardiac Effects of Performance-Enhancing Medications: Caffeine vs. Anabolic Androgenic Steroids"

_diagnostics, 2021, doi:10.3390/diagnostics11020324_

Round 1

Reviewer 1 Report

In this study, the authors addresses the effects of the two commonest PEDs, one legal, the other prohibited, and their respective effects on the heart, as well as the long-term implications.

Comments:

In Introduction section:  the authors  should add more data about anabolic steroids and their side effects such as cardiac complications or even stroke (doi.org/10.3390/jcm8091295)

Line 64, the authors state: “This review follows the guidelines set forth by PRISMA (preferred reporting items for systematic 65 reviews and meta-analyses”; but there is no PRISMA diagram. I strongly suggest to make one and add it!

Line 99-118: a cartoon with mechanism of action will be welcomed.

Line 263: the authors have to present the clinical pitfalls of thie paper, not only the Limitations.

Consider revising accordingly!

Author Response

Manuscript ID: diagnostics-1064607

Dear Professor Kjaer,

We would like to thank the Editors and Reviewers for their interest in our work and for their suggestions and critiques that helped us improve the manuscript. We have revised our manuscript and have incorporated the changes recommended by the Reviewers.

Reviewer 1:

  1. In Introduction section:  the authors  should add more data about anabolic steroids and their side effects such as cardiac complications or even stroke (doi.org/10.3390/jcm8091295)
  • Response 1: We thank the reviewer for this comment, and we have included this in our introduction (Line 69-71). We have expanded further in our Results section (Line 346-351), and have cited the recommended reference.
  1. Line 64, the authors state: “This review follows the guidelines set forth by PRISMA (preferred reporting items for systematic 65 reviews and meta-analyses”; but there is no PRISMA diagram. I strongly suggest to make one and add it!
  • Response 2: We thank the reviewer for this comment, and we have included this in our Appendix (Line 488).
  1. Line 99-118: a cartoon with mechanism of action will be welcomed.
  • Response 3: We thank the reviewer for this comment, and we have included a diagram (Line 133; Figure 1).
  1. Line 263: the authors have to present the clinical pitfalls of thie paper, not only the Limitations.
  • Response 4: We thank the reviewer for this comment, and we have included this in Line 404-416.

Reviewer 2 Report

This review aims to analyze the effects of the two commonest performance-enhancing drugs (PEDs), caffeine vs. anabolic-steroids. Particularly, it aims to focus on the adverse effects on the heart, analyzing the long-term effects. First of all, the hyphenated "Anabolic-androgenic steroids (AAS)" is more conventional and important for indicating that the anabolic and androgenic effects of AAS are inextricable. The “Introduction” section should be improved. To compose this section, the authors have consulted only two references: for a literature review, it is very limited. Moreover, the paragraph about the anabolic-steroids is written without references. Finally, the study’s aims should be improved. Please, check and revisit all section. The section 3 should be split in two sections: Results and Discussion (4). In “results” the two subsections should be improved. Both subsections (Caffeine vs AAS) should be specular: for each substance the same subparagraph should be inserted (_______ as a Performance Enhancing Agent; ______ Pharmacology and Cardiac Physiology; __________ and Risk of Arrhythmia; ____________ Genetics; __________; ___________ in Sudden Cardiac Death). In the section about the anabolic-androgenic substances several important references are missed. I suggest inserting: - 10.3390/medicina56110587; DOI: 10.2174/138920111794295792; DOI: 10.1016/j.forsciint.2009.10.025. In the new section “Discussion” the authors should compare the collected data about caffeine with the collected data about the AAS. In this regard, I believe that it could be useful for the readers to insert a new table summarizing the main finding for each subparagraph. In this section, it could be inserted the limitations. Finally, in the conclusion section, several ideas for future studies should be inserted. For example, I suggest reading 10.3389/fphar.2018.01321, focusing on the cardiovascular system and heart.

Author Response

Manuscript ID: diagnostics-1064607

Dear Professor Kjaer,

We would like to thank the Editors and Reviewers for their interest in our work and for their suggestions and critiques that helped us improve the manuscript. We have revised our manuscript and have incorporated the changes recommended by the Reviewers.

Reviewer 2:

1. First of all, the hyphenated "Anabolic-androgenic steroids (AAS)" is more conventional and important for indicating that the anabolic and androgenic effects of AAS are inextricable.

  • Response 1: We thank the reviewer for this comment, and we have adopted the conventional terminology.

2. The “Introduction” section should be improved. To compose this section, the authors have consulted only two references: for a literature review, it is very limited.

  • Response 2: We thank the reviewer for this comment, and we have expanded our introduction and added further references (Line 54-74).

3. Moreover, the paragraph about the anabolic-steroids is written without references.

  • Response 3: We thank the reviewer for this comment, and we have added references to reflect our findings.

4. Finally, the study’s aims should be improved. Please, check and revisit all section.

  • Response 4: We thank the reviewer for this comment, and we have amended our manuscript to reflect this comment.

5. The section 3 should be split in two sections: Results and Discussion (4). In “results” the two subsections should be improved. Both subsections (Caffeine vs AAS) should be specular: for each substance the same subparagraph should be inserted (_______ as a Performance Enhancing Agent; ______ Pharmacology and Cardiac Physiology; __________ and Risk of Arrhythmia; ____________ Genetics; __________; ___________ in Sudden Cardiac Death).

  • Response 5: We thank the reviewer for this suggestion, and we have split section 3 into Results and Discussion.

6. In the section about the anabolic-androgenic substances several important references are missed. I suggest inserting: - 10.3390/medicina56110587; DOI: 10.2174/138920111794295792; DOI: 10.1016/j.forsciint.2009.10.025.

  • Response 6: We thank the reviewer for this suggestion, and we have included the following references in our manuscript.

7. In the new section “Discussion” the authors should compare the collected data about caffeine with the collected data about the AAS. In this regard, I believe that it could be useful for the readers to insert a new table summarizing the main finding for each subparagraph. In this section, it could be inserted the limitations.

  • Response 7: We thank the reviewer for this suggestion, and we have included a new Discussion paragraph that compares between anabolic-androgenic steroids vs caffeine (Line 358 – 403). We have provided a new figure.

8. Finally, in the conclusion section, several ideas for future studies should be inserted. For example, I suggest reading 10.3389/fphar.2018.01321, focusing on the cardiovascular system and heart.

  • Response 8: We thank the reviewer for this suggestion, and we have included ideas for future studies (Line 420-432).

Yours faithfully,

Dr. Aneil Malhotra

Round 2

Reviewer 1 Report

The authors addressed all the comments of the reviewers. The manuscript is much improved compared with the previous version. I have only a small editorial comment: 

Page 4 line 139 – I think is figure 1 and not figure 2 please correct.

Reviewer 2 Report

Following the reviewers' suggestions, the authors have substantially improved the manuscript. For, this reason, I endorse the publication of the manuscript in its current form.